# Specimen sharing for epidemic preparedness: Building a virtual biorepository system from local governance to global partnerships

**Judith Giri**[1©], **Laura Pezzi**[2‡], **Rodrigo Cachay**[3‡], **Rosa Margarita Gèlvez Ramirez**[4‡], **Adriana Tami**[5¤‡], **Sarah Bethencourt**[6‡], **Anyela Lozano**[7‡], **José Eduardo Gotuzzo Herencia**[3‡], **Julia Poje**[1‡], **Thomas Jaenisch**[1©], **May Chu**[1©]*

1 Center for Global Health, Colorado School of Public Health, Anschutz Medical Center, Aurora, Colorado, United States of America, 2 Unité des Virus Émergents (UVE: Aix-Marseille Univ-IRD 190-Inserm 1207), Marseille, France, 3 Instituto de Medicina Tropical Alexander von Humboldt, Universidad Peruana Cayetano Heredia, San Martín de Porres, Lima, Peru, 4 Centro de Atención y Diagnóstico de Enfemedades Infecciosas-CDI, Fundación INFOVIDA, Bucaramanga, Colombia, 5 Facultad de Ciencias de la Salud, Departamento de Parasitología, Universidad de Carabobo, Valencia, Venezuela, 6 Departamento de Estudios Clínicos-Department of Clinical Studies, Universidad de Carabobo, Valencia, Venezuela, 7 Centro de Investigaciones Epidemiológicas, Universidad Industrial de Santander, Bucamaranga, Colombia

© These authors contributed equally to this work.
¤ Current address: Department of Medical Microbiology & Infection Prevention, University Medical Center, Groningen, Netherlands
‡ LP, RC, RMGR, AT, SB, AL, JEGH, and JP also contributed equally to this work.
* may.chu@cuanschutz.edu

**Data Availability Statement:** Data available at the University of Colorado-Anschutz Campus Digital Collections at https://doi.org/10.25677/qh88-aq74.

## Abstract

We present a framework for a federated, virtual biorepository system (VBS) with locally collected and managed specimens, as a 'global public good' model based on principles of equitable access and benefit sharing. The VBS is intended to facilitate timely access to biological specimens and associated data for outbreak-prone infectious diseases to accelerate the development and evaluation of diagnostics, assess vaccine efficacy, and to support surveillance and research needs. The VBS is aimed to be aligned with the WHO BioHub and other specimen sharing efforts as a force multiplier to meet the needs of strengthening global tools for countering epidemics. The purpose of our initial research is to lay the basis of the collaboration, management and principles of equitable sharing focused on low- and middle-income country partners. Here we report on surveys and interviews undertaken with biorepository-interested parties to better understand needs and barriers for specimen access and share examples from the ZIKAlliance partnership on the governance and operations of locally organized biorepositories.

## Introduction

The continuing and evolving emerging epidemics like Ebola, Zika and COVID-19- to name only a few, highlight the need for timely, transparent and efficient access to quality, specimens with associated data to drive research, development of new diagnostics and treatments, disease surveillance, and to measure vaccine efficacy on a global scale. Accessing quality specimens

**Funding:** The work was partially funded by the European Union Horizon 2020 Research and Innovation Programme under Grant Agreement No. 825746, and is supported by the Canadian Institutes of Health Research, Institute of Genetics (CIHR-IG) under Grant Agreement No. 01886-000.

**Competing interests:** The authors have declared that no competing interests exist.

has been largely driven by pathogen-specific needs and related research interests, with little consideration to the broader sharing of those specimens or collections. Trusted infrastructure and specimen sharing mechanisms need to be in place to support the needs of emergent outbreak responses. Without a supportive infrastructure, equitable sharing of specimens can become a complicated, prolonged and stressful process, blunting the best intentions to quickly respond to public health emergencies [1–6].

Biorepositories (biobanks) focused on biomedical collections, such as those supporting global health and infectious diseases research are a fundamental resource and serve as an essential infrastructure for responsible specimen management [7–10]. There has been great progress in recent years towards improving quality management of different types of institutional biorepositories, based on best practices [11] and accreditation standards- ISO 20387;20189(E) Biotechnology-Biobanking standards (https://www.iso.org/obp/ui/#iso:std:iso:20387:ed-1:v1:en), with robust inventory and data management systems. Many large-scale biomedical biorepositories have been established to meet specimens needs for non-communicable diseases like cancer or genetic disorders and, more recently, to benefit personalized medicine [9, 11, 12]. Large population-based public health and clinical biorepositories have also been established such as the United Kingdom Biobank (https://www.ukbiobank.ac.uk/), the National Health and Nutrition Examination Survey (https://www.cdc.gov/nchs/nhanes/biospecimens/biospecimens.htm), the National Cancer Institute' biorepository (https://www.cancer.gov/research/infrastructure) as well as biobank hubs such as the German Biobank Node [13]. The COVID-19 pandemic has led to a proliferation of collections both in existing and newly established COVID-19 specific biorepositories (PATH, https://www.path.org/programs/diagnostics/washington-covid-19-biorepository/ [14–16]. Yet, despite the existence of the large array of biorepositories, there is still room for additional collections as the existing repositories do not fully represent the diverse range of geography, ecozones, and demographics needed to support emerging infectious disease research response efforts. The need for broader specimen collections for (emerging) infectious diseases, representative of diverse populations, especially from low and middle income countries (LMIC), remains an unmet need even after recent investments in infrastructure during COVID-19, and a transparent sharing system has not been fully realized [5, 6, 14, 17]. A gap beyond our scope and focus, to include the taxonomic diversity of pathogens and reservoirs has also been identified [14].

A major challenge remains the coordination of biorepositories and collections in different geographic areas and overcoming obstacles for access. A few successful models such as the Foundation for Innovative Diagnostics (FIND) DxConnect (https://www.finddx.org/wp-content/uploads/2022/12/20211202_fac_specimen_bank_VF_EN.pdf) and the European Viral Archive-Global (EVAg) that provides a trusted broker role in accessing virus strains, molecular targets, virus components and tissue cell lines [2, 18], and as mentioned in this paper, the ZIKAlliance network of local biorepositories, demonstrate that there are communities of practice willing to share and exchange timely, quality specimens.

A frequently invoked barrier is compliance with the benefit sharing requirement of the Convention for Biological Diversity's Nagoya Protocol (NP, The Nagoya Protocol on Access and Benefit-sharing (cbd.int)). The NP mandates signatories to adopt national requirements for equitable and fair sharing of benefits, including but not exclusive to any benefits from materials that contributed to the development of products such as diagnostic tests, vaccines, or therapeutics. National signatories have adopted differing interpretations of the NP which have become potential barriers for access to specimens [19]. The full impact and benefits enshrined in the NP remains a work in progress and is being actively addressed by the WHO BioHub effort [6].

A serious challenge in sourcing specimens across countries and institutions is the inherent heterogeneity of methods of collection, characterization and handling of specimens and data, and the difficulty this poses for users regarding the quality and integrity of the specimens [20]. This is not surprising, given that maintenance of a biorepository collection is typically an unfunded mandate for a clinic or laboratory. A lack of proactive, coordinated approach leads to inconsistencies and fragmentation of infectious diseases biorepositories and poses challenges to the stakeholder community trying to harmonize biorepository practices and have a need to access specimens. Existing models like FIND and EVAg can serve as exemplars to learn from and guide this project [2, 18].

The abundance of specimens during the COVID-19 pandemic, unlike previous situations in outbreaks where competition for a limited number of specimens was the rule, provided a good opportunity to examine existing gaps for specimen collection, use and sharing [1, 5, 16]. We identified that there are many untapped and interested sites that could be contributors and users of specimens, but we lack a holistic and more cohesive approach to identify and tackle the barriers and foster realistic benefits. An approach, centered on participation of LMIC, who often are at the center of new public health threats or emerging zoonotic hot zones, could prove pivotal in enabling LMIC to lead in collecting and managing specimens for broader sharing [3, 21, 22]. Therefore, we sought to seek input from a broad group with interest in accessing/providing specimens, especially for serological diagnostics, to examine how to best provide practical solutions that utilize their suitable existing infrastructure rather than investing in building large new biorepository facilities. We initiated a series of workshops, and while heavily interrupted by the emergence and spread of COVID-19, the pandemic also provided an opportunity to engage many more who are interested in specimen sharing.

The efforts leading up to this manuscript were initiated at a workshop at the American Society of Tropical Medicine and Hygiene meeting in 2019, co-convened by Center for Global Health at the Colorado School of Public Health and FIND as an outcome of the 2016 Zika virus public health emergency. We have subsequently organized virtual workshops hosted by the Global Health Network (TGHN; https://globalbiorepository.tghn.org/) and (a) initiated a survey and (b) conducted interviews with these goals in mind: to identify the current barriers to specimens sharing and unmet needs of various stakeholders; and to initiate discussion about a proposed distributed, locally-managed virtual biorepository system (VBS) as a solution to meeting the identified challenges.

We based our questions in the survey and the interviews on a vision for a VBS characterized by principles of equitable, open and transparent access to specimens for the 'global public good', and at the same time responding to the needs of diverse stakeholders. We focused on diseases caused by pathogens with outbreak potential and placed emphasis on specimens for development of diagnostics and research. In our survey and interviews we asked participants what their view might be for a sustainable infrastructure focused at the LMIC level that provides benefits to diverse specimen users and providers.

We intend to use a grassroots approach to developing a VBS that will support local governance and stewardship of specimens. Our goal is not to create a comprehensive nor duplicative system but one that works in synergy to complement and fill existing gaps in collections. For these reasons, we sought input from local decision-making and governance structures in LMICs regarding the sharing of biological materials, incorporating longstanding collaborative relationships of the authors (e.g., within the Reconciliation of Cohort Data for Infectious Diseases (ReCoDID), www.recodid.eu; and ZIKAlliance consortia, https://zikalliance.tghn.org/ and AEDES Network, https://www.redaedes.org/). Since 2021, we have joined forces with the

Center for Research in Emerging Infectious Disease (CREID) Coordinating Center (https://creid-network.org/coordinating-center) to guide the activities of the VBS.

## Methods

### Participation and privacy policy

Invitations were extended to those individuals and/or representatives of agencies and entities who had joined previous biorepository activities, to those identified through publications as well as those who had expressed interest through TGHN website. Interested participants agreed to take our online survey while those involved in in vitro diagnostics (IVD) were invited for an interview. Participants were informed that they retained the right to withdraw without penalty and the collected data would be de-identified and reported in aggregate form following the research-ethics guidelines of the American Psychological Society research-ethics guidelines (https://www.apa.org/ethics/code/ethics-code-2017.pdf).

### Survey on benefits in participating in a VBS

An online self-administered 10-question qualitative survey (S1 Table) was created on the SurveyMonkey platform for the participants. The survey was accessed by email or website (https://globalbiorepository.tghn.org/) between December 2020 through January 2022. Data analysis was finalized in December 2022.

### Interviews to assess needs and barriers to accessing specimens

Video-conference interviews were conducted jointly by 2 members of the study team (JG and MC) and recorded with the consent of the interviewees. We targeted potential users of well-characterized specimens, including academic, not for profit and commercial organizations. Commercial users consisted of diagnostics industry companies that represented both small IVD products <10 and mid-size to larger (with IVD products > 10) operations [23]. The companies represent a small portion of the interviewees. They were from Europe, Canada and the US, with the midsize to large companies having international subsidiaries and reach. We interviewed consented participants with regard to their need and use of biological specimens. Our questions were based on their respective strategies in developing diagnostic tests and reference controls and how they would acquire materials for developing an IVD. Interviewees were encouraged to provide additional feedback about specific hurdles they encountered sourcing specimens. The interviews lasted on average from 40 to 60 minutes and responses were compared and tabulated for analysis. The survey is in S2 Table.

### Local biorepository governance structures

We interviewed research teams from the ZIKAlliance consortium [22] including: (a) the Industrial University of Santander, Bucamaranga, Colombia; (b) the University of Carabobo, Valencia, Venezuela; and (c) Cayetano Heredia University in Lima, Peru to learn from their case examples of locally governed biorepositories. Each of these entities established biorepositories in response to Zika virus research needs and were willing to share specimens for advancing research and as reference materials. In Colombia, they already had an established AEDES Network for sharing specimens with network members and we learned how a VBS might also grow through networks. Their collective experiences are used to illustrate their biorepository decision-making governance and operational structures and how this infrastructure may be used to improve locally managed specimen sharing mechanisms.

## Results

### Benefits survey

**Characteristics of survey respondents.**    Forty-seven respondents completed the survey, and their replies were aggregated and analyzed. Respondents logged into the survey from Africa (n = 15), North America (n = 10) Europe (n = 9), Latin America (n = 9) and 5 were from the Asia-Pacific region (Fig 1). They represented a variety of organizations from research institutions (n = 23), not-for-profit or government institutions (n = 22), public health and clinical laboratories (n = 9), biorepositories (n = 5) and commercial diagnostics laboratories (n = 3); a few respondents (n = 4) did not identify their affiliation. Respondents could select more than one category, for example as a research and as a government institution. The participants represent a broad group of stakeholders with a variety of roles, including principal investigators, public health officials, institution leaders, and laboratory or biorepository managers.

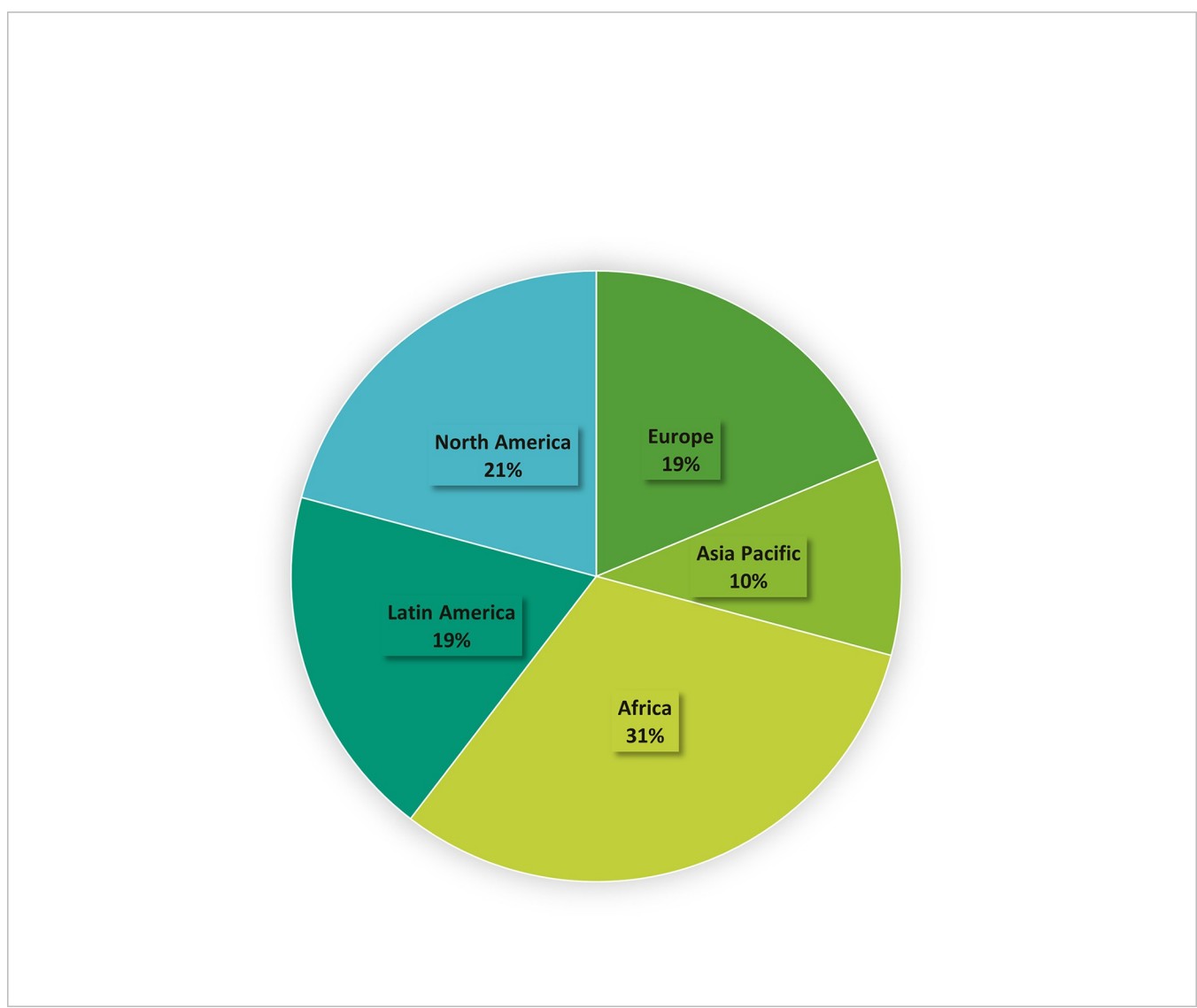

**Fig 1. The geographic location of survey respondents.**

**Benefits of the VBS.** Embedded in our 10-question survey (S1 Table) were 3 questions aimed specifically at understanding what respondents considered to be the key VBS benefits. Survey respondents could choose all that applied to them from a list of potential benefits and rank their importance. These questions were: (a) which planned VBS benefits would be most important to you/your institution? (Q4), (b) which VBS support functions would be most valuable to your current biorepository or collection? (Q5) and (c) which additional resource would allow you to participate in the VBS (Q6).

The highest ranked benefits for investigators or their institutions were: (a) opportunities to network and collaborate with international partners (38/47, 81%), (b) capacity building for infectious disease specimen resources (35/47, 74%), (c) career growth and training opportunities (28/47, 60%), (d) greater visibility and utilization of specimens (26/47, 55%) and (e) additional funding opportunities 25/47, 53%) as shown in Table 1.

When survey respondents were asked to identify which features/functions of the VBS would be of highest benefit to their current or future local biorepository operations (Table 1), the most often selected functions were (a) the availability of standardized procedures (SOPs) and processes including quality standards, templates for Material Transfer Agreements (MTAs) and informed consent procedures, (33/47, 70%); and (b) the coordination of logistics for accessing and distributing specimens, such as communications, review of requests and assistance with legal and ethical issues (30/47, 64%) with (c) inclusion in a catalog and directory and access to laboratory informatics tools and support, were selected as equally valuable features by 24/47 (51%). The feature of a catalogue and directory was the highest ranked valuable VBS function for diagnostics developers, especially smaller companies interviewed, as described in the next section.

In the open comments section, we offered an opportunity to suggest additional desirable benefits that could be provided by the VBS. Comments received stated that the VBS could: (a) serve as forum for sharing successful problem-solving approaches (e.g., legal issues), (b) support start-up of new biorepositories, (c) promote research and development capacity for preparedness in LMICs, and (d) ensure proper attribution for specimen providers and, most importantly, (e) strive for a governance framework in which all partners are treated as equals.

**Barriers to specimen sharing.** A frequently invoked barrier for sharing specimens across borders is a party's compliance with the benefit sharing requirement of the NP. In Q7,

**Table 1. Virtual biorepository functions and services of greatest benefit listed by order of preferences.**

| FOR INVESTIGATORS AND THEIR INSTITUTIONS | N (%) | FOR CURRENT OR FUTURE BIOREPOSITORY OPERATIONAL NEEDS | N (%) |
|---|---|---|---|
| Local and international networking and collaboration | 38 (81) | Standardized procedures and processes and templates (SOP, quality, MTA, consent) | 33 (70) |
| Capacity building for infectious disease biorepositories | 35 (75) | Coordination of access and distribution of specimens (communication, review of requests, logistics and legal/ethical) | 30 (64) |
| Career growth and training opportunities | 28 (60) | Inclusion in online catalog or directory specimen resources | 24 (51) |
| New funding opportunities | 25 (53) | Laboratory Informatics tools and support | 24 (51) |
| Reputation as trusted source of Specimens | 22 (47) | Access to biorepository expertise/collections management | 22 (47) |
| Supporting test developers and industry | 20 (43) | Coordination of laboratory services | 13 (28) |
| Authorship on publications | 18 (38) | | |
| Other comments | 4(8) | Other comments | 5 (11) |

respondents were asked, if their country had any policies in accordance with the NP and whether it would prevent them from contributing specimens under the VBS. An interesting, but not surprising result was that more than half of the responses (53%, 25 out of 47) indicated lack of familiarity with their own country's status regarding the NP and the respective impact on specimen sharing. Respondents from countries with specific regulations in accordance with the NP were asked to elaborate whether the rules would prevent sharing outside the country. Some (19%, n = 9) indicated that sharing was possible with approval from appropriate authoritative officials, such as their respective Ministries of Health. When international sharing was restricted, some believed they could still participate and use their local specimens to test in-country, if reagents or kits were provided.

**Other findings.** We asked respondents about the feasibility of contributing a set of qualified specimens from their COVID-19 specimen collections to the VBS to create evaluation panels, and 16 /40 of the responses (40%) indicated that they could provide such specimens. We will plan to follow up and explore whether the VBS could provide specimen panels as a service. Several additional items for follow up were indicated in comments (S1 Table, Q10) including: (a) barriers to sharing such as shipping of specimens; (b) recommendations for a successful partnership to be based on equity, respect and credit for contribution, (c) support for a VBS as a catalyst for research collaborations and, (d) the benefit of the VBS providing access to biorepository expertise.

Finally, 40/47 respondents replied via comments to Q8 (S1 Table) which asked them if they would be inclined to participate in a VBS and why they would join: 29/40 indicated they would join and the reasons were about specimens, research, networking and an opportunity for benefits, affirming the choices in Q4-Q6 (S1 Table).

## Interview results with specimen users and providers: Identifying unmet needs and barriers to access

**Characteristics of interviewees.** We specifically invited representative organizations with different perspectives and needs for detailed interviews: (a) developers of IVDs of differing sizes and likely interested specimen users and (b) organizations providing specimens and related services, reference specimen providers, and managers of networks of collections.

A total of 11 interviews were conducted. We analyzed their responses in two distinct groups (1) 4 small companies and 2 large/midsize for-profit enterprises with global markets; and (2) 5 not-for-profit enterprises, including an external quality assessment provider and interviewees from academia with government support or seconded to, government organizations. Despite the small number of interviewees, we were able to capture a wide and diverse range of key insights.

**Gaps and barriers for access to specimens.** Among those interviewed, we found that their priorities varied, however, they shared common gaps and barriers, summarized in Table 2.

A detailed summary of the interview results is included in the S2 Table and demonstrates the common concerns as well as differences in the types of barriers and hurdles of highest

**Table 2. Common gaps and barriers of high concern for access to specimens.**

a) Timely access to characterized high quality specimens
b) Availability of specimen types needed for pathogen detection and test development; longitudinal collections and specimens collected at different times in an outbreak (before, during and after)
c) Access to reliable and complete accompanying information (clinical data; specimen handling conditions)
d) Need for specimen panels and reference materials for quality assessment and validation

concern for different types of organization interviewed. Availability of some specimen types in the context of COVID-19, but especially for the previous outbreaks (limited geographically, seasonally, recurrence, etc.) was a universally recognized concern by all interviewees.

Some differences in ranking of the barriers were also evident from the interviews (S2 Table), depending on the type of organization represented. For commercial IVD developers, a comparison of larger vs small IVD developers indicated that for smaller companies the identification of reliable trusted sources of specimens were a very high priority for accelerating time to development and approval of new diagnostics. Interviews indicated that such companies need to identify and negotiate with new sources of specimens for each new pathogen or obtain specimens from commercial sources, while larger companies are more likely to have long-standing collaborations with existing legal agreements. A related concern was access to reliable and sufficient data to meet regulatory requirements, especially for commercially acquired specimens or use of left-over specimens. where control and oversight are less strictly managed. Data access restrictions, especially in European countries due to privacy rules, impacted access to specimen associated information for all users. The cost of acquiring specimens was also cited as a significant financial burden, especially when the specimens obtained were of poor quality and not fit-for-purpose. These issues affected all types of organizations but appeared to disproportionately impact small companies especially early in outbreaks, when timeliness and efficiency are most critical, resulting in considerable delays in development and validation of new tests.

Organizations that work in the international sphere, primarily the not-for-profit organizations that broker specimens for research and for commercial and other entities, identified the NP and related policies as the greatest barriers to access. They also expressed concern about the complexity of import and export controls, disparities in regulations and inconsistencies between countries regarding biospecimens and biosafety/biosecurity policies, that can delay efficient and timely access to specimens. Lack of trust was identified as another barrier for accessing specimens from LMICs, including concerns about ethical use of specimens, and communicating that biorepositories need investment in capacity building and training for partners from LMICs. This group also provided specific recommendations regarding features of the VBS that in their experience would meet the needs of various stakeholders (Table 3), specifically these recommendations aimed to reduce the complexities around specimen access and specimen sharing under an operational structure that can maintain quality and confidence in the materials to be accessed.

## Local governance models

During the Zika epidemic in South America (2016), ZIKAlliance partner institutions developed a common governance structure for specimen sharing, to support research studies (Fig 2). To learn how the VBS could facilitate sharing of specimens and what hurdles may be encountered, we examined how in-country and international transfer of specimens was

**Table 3. Recommendations regarding features of a VBS.**

a) Setting criteria for specimen and data quality that are deemed appropriate for inclusion into the VBS, with a potential grading to indicate fit for different uses
b) Offering access to software to some users as needed to facilitate data sharing and resolve the hurdles created by use of disparate platforms and applications
c) Standardizing of MTAs and negotiations to facilitate sharing and to enable agile responses in cases of public health emergencies
d) Emphasis on quality to including not only specimen and data but also biorepository operations, such as standards for managing and preserving specimens
e) Facilitation of access with a simplified review process by the VBS
f) Building biobanking capacity and trust in LMICs to enable equitable access to specimens and benefits

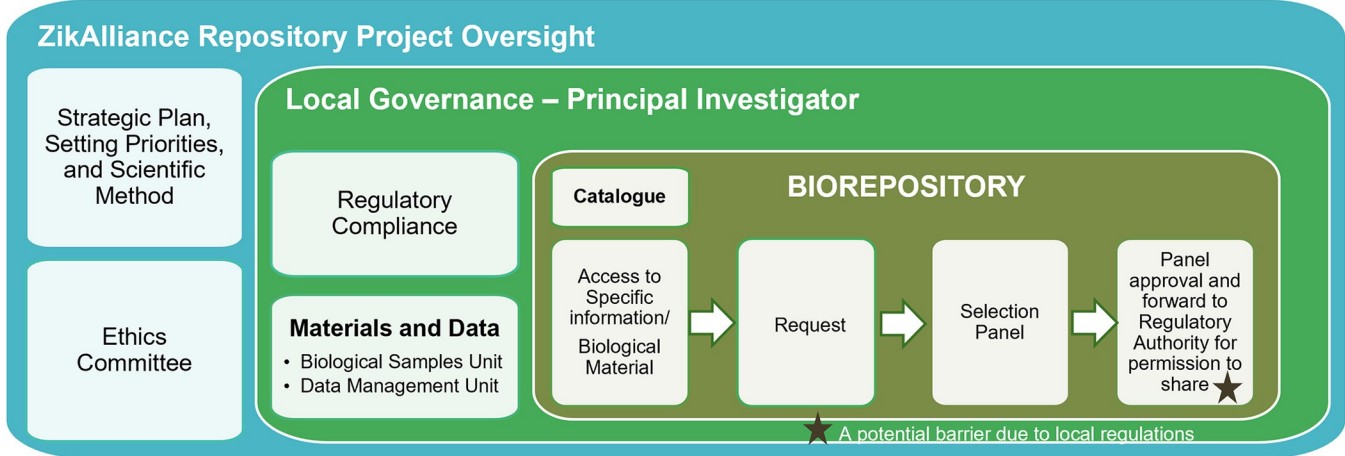

**Fig 2. Governance structure of the ZIKAlliance local biorepositories.**

accomplished among ZIKAlliance partner institutions. The ZikaAlliance framework developed the overarching ethics and research plan, but decision about access to specimens resided with the local principal investigator (PI) and the local Selection Panel (SP). Approved requests were forwarded to the regulatory agencies, which in Venezuela and Colombia is the Ministry of Health and Social Protection and in Peru the Ministry of Health (S1A–S1C Fig).

For transfer specimens within the ZIKAllinace outside of the country, for example from Colombia to a reference laboratory in France, in addition to local ethics board approval, a permission to share was required from the Ministry of Health and Social Protection of Colombia (https://www.minsalud.gov.co/Paginas/default.aspx). The approval process, from time of submission to official notification of the PI by the Ministry typically took about 1–2 months.

Another example linked to this effort is the AEDES network in Colombia, where a Scientific Management Committee is responsible for coordinating and monitoring the progress of the technical, scientific and institutional components of the network in accordance with its Scientific Advisory Board and their Institutional Review Board or Ethics Committee.

As illustrated by local governance models (Fig 2 and S1A–S1C Fig) a publicly accessible catalogue of the biological specimens available for sharing was the key starting point (Fig 2). The governance structure, allowed for a careful, thoughtful evaluation process for requests by the local principal investigator (PI) and the process also took into account the need for recruitment and replenishment of specimens. A future challenge beyond ZIKAlliance and similar models will be the availability of funds for sustainability, and scientific resolve to grow and maintain local biorepositories and their governance systems.

## Discussion

### Benefits survey results and its implications

Overall, our results showed that networking opportunities and capacity building for infectious disease biorepositories were considered of high importance, and this was confirmed through open survey comments. The responses suggested areas for further follow up that we intend to pursue through a Delphi-prioritization exercise. In order to operationalize the VBS, we would like greater clarity on what could constitute capacity building, the implications of the NP on specimen sharing and methods different countries adopt to receive the benefits [19].

The VBS is aimed to work with other biorepository efforts particularly those focused for LMICs to collaboratively meet a global gap. Based on the information we gathered regarding the need for coordination, a key feature of the VBS would be its role as 'trusted broker' or "navigator" for ensuring the source of high-quality specimens for future pandemic threat agents, which includes the availability of more granular information on the clinical phenotype linked to the specimen and the potential for sequential specimens over the course of the illness. The broker status would also include enhanced accessibility due to successful navigation of legal and logistics hurdles for which good examples already exist.

A future focus is the use of the VBS as a source of specimens for reference materials and panels. For example, one of the models considered for the VBS, is to request specimen providers to identify and set aside a few large volume well-characterized high-quality specimens that can be accessed through the VBS, for use for evaluation of reference panels needed for calibration of diagnostics; this would require setting aside only a portion of their respective collections rather than their entire biorepository or specimen collection for a pathogen of interest. About 40% of respondents indicated that they would consider setting aside a small set of qualified specimens from their COVID-19 specimen collections for sharing within the VBS.

## Interviews with specimen users: Identifying unmet needs and barriers to access

In the interviews, many of the participants from various types of organizations voiced the hope that the features of the VBS would provide solutions to barriers identified, differing primarily in ranking of priorities based on the types of organizations and their mission, such as small compared to large established diagnostics developers and the needs of research entities. The key features of this VBS identified by all were: (a) serving as a trusted, reliable resource for identifying and accessing characterized specimens and associated data and, (b) availability of characterized specimens of sufficient volume and quality fit for evaluation panels. Characterization of the specimens in a standardized process will benefit partners especially in LMICs where there is less opportunity to conduct the characterization themselves, thus providing them a resource that could allow them to develop diagnostics locally. An example of a benefit that the respondents identified is the possibility of cross collaboration to create clinical evaluation panels to validate and utilize for license submission for diagnostic test kits, especially where public health and test developers can provide access to clinically confirmed case specimens that would be difficult to assemble by a single entity, such as the comprehensive panel of sera from patients with various states of Lyme disease, and healthy persons for Lyme disease serological diagnostic testing [24].

## Comparison of local governance models

The local governance examples were considered as models for local management of specimens in a VBS. Applications for specimens would be reviewed by the consortium representatives, but the institution would make the decision for sharing and retain local control of the specimens.

By promoting local biorepositories in LMICs, we can facilitate technology transfer opportunities and capacity building even if dedicated funding cannot be guaranteed. We will emphasize fit-for-purpose technologies appropriate for LMIC settings (lyophilized and dried stocks, for example, instead of low temperature freezing). We must adhere to transparency of processes and ethical standards in order to prevent misuse of specimens as well as avoiding any perception of exploitation for specimen providers and users alike. The focus on local biorepositories echoes the need expressed in the interviews for help with the operational challenges,

including availability of harmonized platforms and logistics. This can be implemented by formal agreements between VBS partners in the system, and the selection of partners based on willingness to adhere to the governance criteria (specimens, data, biorepository operations, MTA, sharing etc.) will. Such services require consistent monitoring, that adapts to all stakeholders and specifically, the VBS specimen user's needs.

## Limitations of this study

The survey was delayed from conception in 2019 to completion in 2021, interrupted by the on-going COVID-19 pandemic. We made attempts to enroll all who had expressed interest through our workshops to participate in our survey and interview on the concept of a VBS. All the respondents expressed interest in solving existing gaps for specimens' access, but not everyone had a complete or in-depth understanding of all the issues around sharing or the requirements of maintaining a non-centralized, federated biorepository. An outcome of this first survey is to understand the type of engagement we might expect in a proposed VBS operation, and we are actively pursuing steps to operationalize the proposed VBS.

## Conclusions

Centralized biorepositories tend to require major investments in infrastructure with committed long-term support. Due to the resources required to maintain such facilities, these types of biorepositories are primarily established in higher income countries. For infectious diseases research, this approach does not favor biorepositories in places where new pathogens may emerge. There is a global unmet need for availability and access to quality specimens across a diversity of locations from which disease "X" may emerge. In the absence of specimens from low resource settings, laboratory tests and vaccines may be developed without the opportunity to fully assess their performance in all populations.

The vast spread of the COVID-19 pandemic and its duration has provided an abundance of specimens. More than 100 sites found in a global search have identified themselves as COVID biobanks [1, 14, 16, 20, 25]. Most are uncertain about their long term future, and this dilemma may offer an excellent opportunity to address challenges for sustainability, access and sharing.

In our study we included the perspective of the specimen users who are interested in high quality specimens related to infections with outbreak potential. This allowed us from the specimen collection perspective to examine their needs and identify the barriers in sourcing specimens along with understanding the types of equitable benefits that participants themselves positively identify as acceptable. We examined what might constitute a fair exchange between the investigators collecting well-characterized specimens in LMIC and specimen users; a system in which specimens of high value can be collected and replenished while benefits of recognition and research participation activities can be fostered.

In recent years, recognizing the lack of knowledge of types of specimens that may be available and lack of access to these specimens, multiple consortia have been able to successfully provide information and materials through their operations, however they can't cover the entire spectrum of needs [1, 5, 17], We propose that this is where a VBS model can align and contribute. We especially recognize that there is a need for well-annotated collections of serum/plasma from diverse geographical and disease exposure experiences [14, 20]. A clear focus on this specimen type affords us the opportunity to build the required infrastructure and activities of the VBS as proof of concept.

We acknowledge that national laws and recommendations can preclude exporting specimens, however partners in these areas remain open to collaborative research. In these

circumstances, diagnostic kits and protocols might be shared with reference calibration materials to the site for evaluation, thus setting the scene for active research collaboration.

Biorepositories serve many functions, and there is no single approach or model that fits all needs [7, 20, 26, 27]. We intend to focus on biorepository collections for infectious diseases in global preparedness by enabling and accelerating research and development of interventions through rapid and diverse sharing of human specimens to inform outbreak control strategies. Our concept of creating a distributed, grassroots biorepository is in alignment with concurrent efforts of the DxConnect and is modeled after the successful virus exchange by EVAg [2]. Our effort through TGHN and partnership with ZIKAlliance, ReCoDID and CREID allows us to fill in gaps in collection from different human populations, from geographically diverse eco-zones and human exposomes. A virtual biorepository that functions as a trusted broker under a federated system but without a more resource intensive centralized facility, with specimens shared from local sites as needed is a nimble approach that allows many to participate without high costs. A similar disseminated system was proposed for natural history biorepositories [14]. The proposed VBS would also allow local generators of specimens to receive benefits for setting aside potentially high-value specimens. Sets of well-characterized and richly annotated specimens can be queried in a database under a common governance framework will allow contributors and users alike to access materials respectfully as equal partners. Recent post-public health emergency epidemic reviews on lessons-learned for the future, note preparedness must include access to data and specimens [2, 6, 28–31]. Currently support for biorepository functions remain as a largely unfunded mandate for many, and the recommended actions can raise champions to promote and socialize the benefits of specimen access and encourage investment in the sharing systems.

In this paper, we present arguments for building a trusted specimen sharing system based on a distributed stewardship at the local level, to manage access to quality, well-characterized specimens for outbreak-prone infectious diseases as a global public good. Accessing specimens, even within established networks and projects, remains fragmented with procedures that are lacking transparency and hindered by barriers leading to long lag time and high costs. We obtained knowledge, insights and advice through our workshops, questionnaires, interviews and case examples from public health experts, infectious disease researchers, biorepository managers, clinicians, policy makers, regulators, and diagnostic industry representatives to ascertain what might be appropriate entry points to better organize fair and equitable specimen sharing experience, that allows more engagement with source providers and stimulate research. The following challenges remain: (a) generating quality specimen collections, (b) overcoming barriers to accessing specimens (regulatory, cost, sharing mechanisms), (c) creating a supportive exchange infrastructure, (d) defining benefit packages, and finally, (e) maintaining trust between parties. All these add up to requiring the seeker of specimens to have persistence, commitment of precious time and financial support as the outbreak of interest slips away, only to repeat this cycle again for the next outbreak. We propose a coordinated approach with logistics support and assistance with procedures to facilitate exchange between specimen providers and user. This proposed system may work well and complement existing biorepositories efforts to provide an alternate way to shorten the time to obtain results for decision making for preparedness and response to future public health emergencies.

Our VBS vision is one that prioritizes respect for contributors and users of biological materials. The concepts of equitable benefits sharing, and capacity building will be essential for our effort to be successful and durable. And finally, a coalition of the willing is key to ensure funding, infrastructure and services to realize our vision. We look forward to formalizing the framework and operationalizing the VBS globally.

## Supporting information

**S1 Table. Virtual biorepository benefits questionnaire.**
(DOCX)

**S2 Table. Interviews to assess needs and barriers to accessing specimens.**
(DOCX)

**S3 Table. Interview summary of gaps and barriers to access.**
(DOCX)

**S1 Fig.** A. The Industrial University of Santander, Bucamaranga, Colombia and the Fundación INFOVIDA and Centro de Atención y Diagnóstico de Enfermedades Infecciosas-CDI. B Cayetano Heredia University, Lima, Peru. C. University of Carabobo, Valencia, Venezuela.
(DOCX)

## Acknowledgments

The work was partially funded by the European Union Horizon 2020 Research and Innovation Programme under Grant Agreement No. 825746. We are indebted to FIND (Stefano Ongarello and Dominique Allen) for co-sponsoring our first workshop and for joint discussions with PATH (Roger Peck and Helen Storey) on the concept of the VBS. We thank Dr. Amy Price of Stanford University for her editorial and conceptual inputs. We are grateful to Zainab Al-Rawni, Adam Dale and Trudie Lang of TGHN for hosting our VBS website and access to the workshops.

## Author Contributions

**Conceptualization:** Judith Giri, Thomas Jaenisch, May Chu.

**Data curation:** Julia Poje.

**Formal analysis:** Judith Giri.

**Methodology:** Judith Giri, Thomas Jaenisch, May Chu.

**Project administration:** Julia Poje.

**Supervision:** Thomas Jaenisch, May Chu.

**Visualization:** Laura Pezzi, Rodrigo Cachay, Rosa Margarita Gèlvez Ramirez, Adriana Tami.

**Writing – original draft:** Judith Giri, Julia Poje, Thomas Jaenisch, May Chu.

**Writing – review & editing:** Laura Pezzi, Rodrigo Cachay, Rosa Margarita Gèlvez Ramirez, Adriana Tami, Sarah Bethencourt, Anyela Lozano, José Eduardo Gotuzzo Herencia, Julia Poje, Thomas Jaenisch.

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
