## [Decision Letter · Decision Letter 0]

14 Apr 2023

PGPH-D-22-02118

Specimen sharing for epidemic preparedness: building a local-to-global virtual biorepository framework

Dear Dr. Chu,

Thank you for submitting your manuscript to PLOS Global Public Health. After careful consideration, we feel that it has merit but does not fully meet PLOS Global Public Health’s publication criteria as it currently stands. Therefore, we invite you to submit a revised version of the manuscript that addresses the points raised during the review process.

We look forward to receiving your revised manuscript.

Kind regards,

Syed Shahid Abbas, MBBS, MPH, Ph.D.

Academic Editor

Journal Requirements:

Additional Editor Comments (if provided):

Reviewers' comments:

Reviewer's Responses to Questions

**Comments to the Author**

1. Does this manuscript meet PLOS Global Public Health’s publication criteria? Is the manuscript technically sound, and do the data support the conclusions? The manuscript must describe methodologically and ethically rigorous research with conclusions that are appropriately drawn based on the data presented.

Reviewer #1: Yes

Reviewer #2: Partly

Reviewer #3: Yes

2. Has the statistical analysis been performed appropriately and rigorously?

Reviewer #1: N/A

Reviewer #2: N/A

Reviewer #3: I don't know

3. Have the authors made all data underlying the findings in their manuscript fully available (please refer to the Data Availability Statement at the start of the manuscript PDF file)?

Reviewer #1: Yes

Reviewer #2: Yes

Reviewer #3: Yes

4. Is the manuscript presented in an intelligible fashion and written in standard English?

Reviewer #1: Yes

Reviewer #2: No

Reviewer #3: Yes

5. Review Comments to the Author

Reviewer #1: Dear authors,

compliments for your study. I think it is very useful for the scientific community.

To report only a few punctuation and typing errors.

Line 402: "persons .for Lyme disease serological diagnostic testing" comma not full stop.

Line 442 and 443: dashes

Line 475: after "exposomes" there is an extra space.

Regards

Reviewer #2: The authors propose a Virtual Biorepository System (VBS) as a potential solution to global specimen sharing and epidemic/pandemic responsiveness. I appreciate the goals and sentiment and their call for increased support of biorepositories particularly in LMICs, but overall, I found the perspective narrow and did not see how the VBS framework differs from existing tools and protocols. For example, lines (L) 327-332 exactly describe the formal cataloging process, public databasing, and specimen loan procedures of natural history museums, but with no reference to that community or their published SOPs/best practices. I encourage the authors to expand the perspective to the wealth of literature emanating from non-human biorepositories, see: Bakker et al. 2019, Galbreath et al. 2019, Phillips et al. 2019, Brooks et al. 2020, Colella et al. 2020, NASEM 2021, Theirs et al. 2021, Thompson et al. 2021, Adler Miserendino et al. 2022, Hoberg et al. 2022, among others. Last, it’s unclear how (or if) VBS will plug into existing biorepository database infrastructure (e.g., Arctos, Specify, Symbiota), established data standards (e.g., DarwinCore), recent efforts to inventory global tissue resources (e.g. GGBN), or if VBS intends to reinvent the wheel?

Major

The manuscript almost exclusively addresses commercial uses of biological materials and I wish the authors were more candid about that focus in the abstract/intro, possibly even in the title. Otherwise, how will non-commercial research be accommodated? For example, framing biological materials within ‘a federated system [based] on a demand-and supply model’ (L476) may make sense for commercial establishments, but it positions biological samples as a commodity that is then bartered and traded by governments. Such a model unintentionally excludes non-commercial researchers and organizations (e.g., universities), particularly those from LMICs, from participation due to ‘buy in’ requirements. How will this model be any different?

Historical reliance on serology has limited our ability to extend and/or validate early EID investigations and, critically, ignores accurate taxonomic identifications of hosts. Can the authors rationalize the focus on serological testing methods in this paper, as opposed to shifting to a more holistic sampling approach that would enable more diverse research applications (e.g., genomics, transcriptomics, isotopic work, contaminants, etc.)?

Interviewing only ZIKAlliance members (L163) regarding governmental structure is a serious bias that needs to be addressed. Further, how were the surveys disseminated? Of those targeted, how many responded? How does the inclusion of ‘unidentified’ affiliations affect the results or did unidentified individuals also always select one of the other categories?

The narrow view of biorepositories defined in L431-434 directly contradicts L467 (‘Biorepositories are of many types and functions’). Perhaps, there is an opportunity at this point in the manuscript to contrast the narrow biomedical(?) definition of biorepositories against the broader definition that also includes natural history biorepositories.

Minor

There are grammatical issues throughout the manuscript, please review (e.g., L65-67, L316, L367, L394, L402, L424).

L16: Define ‘annotated’ specimens

L23: What kind of barriers?

L32: Add Philips et al. 2019 as an example of best practices, particularly for cryogenic tissue collections. American Society of Mammalogist’s Systematic Collection Committee is another accrediting institution.

L34-36: Has this been quantified? There are also very large natural history biorepositories (100s of thousands to millions of samples/specimens) developed for other purposes that are also available for disease research.

L62-63: All biological materials are subject to the Nagoya Protocol, not just those that contribute ‘to the development of products’.

Tables: All 3 tables seem unnecessary. For Table 1, all values are reported in the text. Tables 2-3 only have a single column and could be easily summarized within the main body of the manuscript. Note, the word ‘consortium’ has a line through it in Table 3.

L101: “for the ‘global good’” has been used by commercial enterprises to rationalize resource exploitation of LMICs and is why the Nagoya Protocol exists. Please rephrase.

L404-418 echo the goals of the ProjectECHO Museums and Emerging Pathogens in the Americas (MEPA) Community of Practice. Consider citing Colella et al. 2021

L418: remove ‘specifically’

L466: Are all ‘clients’ in this context commercial? Not the case at universities, federal agencies, and NGOs.

Figure 1 does not add the main message of the manuscript – all values are reported in text.

SI Figure 1A-C text need to be enlarged. What do the colors mean?

Suggested references

Adler Miserendino et al. (2022). The Case for Community Self-Governance on Access and Benefit Sharing of Digital Sequence Information. BioScience, 72(5), 405-408.

Bakker et al. (2020). The Global Museum: natural history collections and the future of evolutionary science and public education. PeerJ, 8, e8225.

Brooks et al. (2020). Before the pandemic ends: making sure this never happens again.

Colella et al. (2021). Leveraging natural history biorepositories as a global, decentralized, pathogen surveillance network. PLoS Pathogens, 17(6), e1009583.

Galbreath et al. (2019). Building an integrated infrastructure for exploring biodiversity: field collections and archives of mammals and parasites. Journal of Mammalogy, 100(2), 382-393.

Hoberg et al. (2022). The DAMA Protocol, an Introduction: Finding Pathogens before They Find Us.

Phillips et al. (2019). Curatorial guidelines and standards of the American Society of Mammalogists for collections of genetic resources. Journal of Mammalogy, 100(5), 1690-1694.

National Academies of Sciences, Engineering, and Medicine. (2021). Biological collections: Ensuring critical research and education for the 21st century. National Academies Press.

Thiers et al. (2021). Implementing a community vision for the future of biodiversity collections. BioScience, 71(6), 561-563.

Thompson et al. (2021). Preserve a voucher specimen! The critical need for integrating natural history collections in infectious disease studies. Mbio, 12(1), e02698-20.

Reviewer #3: The manuscript is a qualitative research article developed from the analysis of products of interviews, workshops, and survey/questionnaires on proposed Virtual Biorepository System (VBS) framework for specimen sharing for outbreaks preparedness. Generally, the sample size ie number of individuals, institutions, organizations and countries' opinion sampled were low to support effectively the authors' objective of establishing the VBS framework for specimen sharing.

The objective is noble and seriously needed to strengthen the global fight against future outbreaks, as such this work can serve as foundation for further research on VBS, which the researchers would recruit, collect and analyze larger and more complex data from more individuals, institutions, organizations and countries to develop the VBS framework. The following were some other review comments for the authors attention:

On line 48 on the Manuscript, LIMC Acronyms was used for the first time without writing it in full.

On line 68- 70, a categorical statement was made without any reference being cited.

On line 92, ASTHM acronyms was also used for the first time without writing it out full.

On line 149, the methodology of the study left out questions around opinion of respondents on VBS as it affects their countries' national biosecurity.

On line 152 - 153, Categories of diagnostics industry companies were used in the paper without citing any reference.

On line 154, the geographical coverage of the entities interviewed was not clear. Was their coverage global or restricted. This could skew findings/opinion and render it not representative enough.

On line 178, Don’t you think this number of respondents (47) is too low to power the deductions from the survey, especially as they also have wide characteristic base.

Throughout this paper, the authors chose to use a mix of the word Specimen and Sample in the manuscript – will it not be better to stick to one nomenclature, unless each word carries different meaning where it was used which may be difficult to proof in this paper.

On line 265, Market size - Can you insert reference for the classification of the in vitro diagnostics into the different market sizes used in this manuscript.

On line 301, correct the "European counties" in the manuscript to "European countries".

On line 320, Consortium was cancelled - Was it not currently in place or Is it a suggestion for future implementation.

On line 332, The following statement is missing some words, as such it's not reading well - "the PI the forwards it for to the next level to the regulator".

Thank you.

6. PLOS authors have the option to publish the peer review history of their article (what does this mean?). If published, this will include your full peer review and any attached files.

**Do you want your identity to be public for this peer review?** For information about this choice, including consent withdrawal, please see our Privacy Policy.

Reviewer #1: No

Reviewer #2: No

Reviewer #3: No

---

## [Decision Letter · Decision Letter 1]

5 Sep 2023

Specimen sharing for epidemic preparedness: building a local-to-global virtual biorepository framework

PGPH-D-22-02118R1

Dear Dr. Chu,

We are pleased to inform you that your manuscript 'Specimen sharing for epidemic preparedness: building a local-to-global virtual biorepository framework' has been provisionally accepted for publication in PLOS Global Public Health.

I apologise for the time taken for the peer review process. While I found your manuscript to be well written, and addressing a clearly important topic, it took a lot of effort to find reviewers who were qualified as well as available.

Best regards,

Syed Shahid Abbas, MBBS, MPH, Ph.D.

Academic Editor

Reviewer Comments (if any, and for reference):

Reviewer's Responses to Questions

**Comments to the Author**

1. If the authors have adequately addressed your comments raised in a previous round of review and you feel that this manuscript is now acceptable for publication, you may indicate that here to bypass the “Comments to the Author” section, enter your conflict of interest statement in the “Confidential to Editor” section, and submit your "Accept" recommendation.

Reviewer #3: All comments have been addressed

Reviewer #4: All comments have been addressed

Reviewer #5: All comments have been addressed

2. Does this manuscript meet PLOS Global Public Health’s publication criteria? Is the manuscript technically sound, and do the data support the conclusions? The manuscript must describe methodologically and ethically rigorous research with conclusions that are appropriately drawn based on the data presented.

Reviewer #3: Yes

Reviewer #4: Yes

Reviewer #5: Yes

3. Has the statistical analysis been performed appropriately and rigorously?

Reviewer #3: N/A

Reviewer #4: Yes

Reviewer #5: Yes

4. Have the authors made all data underlying the findings in their manuscript fully available (please refer to the Data Availability Statement at the start of the manuscript PDF file)?

Reviewer #3: Yes

Reviewer #4: Yes

Reviewer #5: Yes

5. Is the manuscript presented in an intelligible fashion and written in standard English?

Reviewer #3: Yes

Reviewer #4: Yes

Reviewer #5: Yes

6. Review Comments to the Author

Reviewer #3: The revised manuscript is richer in scientific content and read well. However, kindly address the observed hanging word "will" on line 393 of the revised manuscript.

Reviewer #4: Since these survey was self administered, you haven't addressed how to control for double reporting by the participants.

Reviewer #5: I think comments from the previous reviewers were well addressed and authors updated the manuscript as suggested wherever possible. Manuscript addresses the objective of the research study and will definitely be very useful for VBS in future for preventing and/or battling with outbreaks.

The only question I have is authors mentioned about interviewing only academicians, not for profit and commercial organizations, but I think views from public health experts, epidemiologists specifically from Government side could have added more value to the findings which I think only responded through survey questionnaire.

7. PLOS authors have the option to publish the peer review history of their article (what does this mean?). If published, this will include your full peer review and any attached files.

**Do you want your identity to be public for this peer review?** For information about this choice, including consent withdrawal, please see our Privacy Policy.

Reviewer #3: No

Reviewer #4: No

Reviewer #5: **Yes: **Dr. Shrikant Kalaskar
